# Validation of new tablet-based problem-solving tasks in primary school students

**Jonas Schäfer**[1,2]*, **Timo Reuter**[2], **Miriam Leuchter**[2], **Julia Karbach**[1,3]

**1** Department of Psychology, University of Kaiserslautern-Landau, Landau, Germany, **2** Institute for Child and Youth Education, University of Kaiserslautern-Landau, Landau, Germany, **3** Center for Research on Individual Development and Adaptive Education of Children at Risk (IDeA), Frankfurt, Germany

* jonas.schaefer@rptu.de

**Data Availability Statement:** The data that support the findings of this study are openly available from the GitHub database in the repository GTT_Validation under the following URL: https://github.com/jonato-bit/GTT_Validation.git.

## Abstract

Problem-solving is an important skill that is associated with reasoning abilities, action control and academic success. Nevertheless, empirical evidence on cognitive correlates of problem-solving performance in childhood is limited. Appropriate assessment tools are scarce and existing analog tasks require extensive coding. Thus, we developed and validated new tablet-based versions of existing analog tasks assessing technical problem-solving with gear construction tasks. To validate these tasks, 215 children (6–8 years) performed the problem-solving tasks in both modalities (analog, digital). To investigate whether performances in both modalities were correlated with other cognitive abilities, participants performed three additional tasks assessing language, reasoning and problem-solving. Structural equation modelling showed that performance was substantially correlated across modalities and also correlated with language, reasoning and another problem-solving task, showing the convergent validity of the digital tasks. We also found scalar measurement invariance across task modalities indicating that both task versions can be used interchangeably. We conclude that both versions (analog and digital) draw on similar cognitive resources and abilities. The analog tasks were thus successfully transferred to a digital platform. The new tasks offer the immense benefits of digital data collection, provide a valid measuring tool advancing problem-solving research in childhood and facilitate the application in the field, e.g., in the classroom.

## Introduction

Problem-solving refers to the process of achieving a goal state that is different from an initial state by performing a series of cognitive or motor actions [1]. This process is typically characterized by the following interdependent phases: (a) understanding and mentally representing the problem, (b) developing plans and strategies to solve it, (c) practically implementing the plans and (d) evaluating (intermediate) outcomes [2]. Aside from minor variations, there is much agreement on these four phases of problem-solving across different domains [3–5]. Problem-solving skills develop significantly during the pre- and primary school years and benefit many important life outcomes, e.g., academic success and social skills [6]. Understanding,

**Funding:** This work was supported by the Ministry of Science, Further Education, Research and Culture, Rhineland-Palatinate, Germany. The funders awarded ML and JK (without grant number). The funders had no role in study design, data collection and analysis, decision to publish, or preparation of the manuscript.

**Competing interests:** The authors have declared that no competing interests exist.

structuring and intentionally solving a problem requires perceptual, motor and creative resources [7]. Exposure to problem-solving challenges promotes reasoning and learning processes in children [8] and is associated with systematic and critical thinking [9]. Therefore, problem-solving abilities are considered major educational learning outcomes [4]. Furthermore, problem-solving is closely related to higher-order cognitive functions, such as fluid intelligence [10] and executive functions [11].

*Technical problem-solving* is considered a prototypical subtype of general problem-solving [4], since it implements the four phases of problem-solving by performing manual actions (phase c) with observable effects (phase d) that are clearly different from the cognitively-based planning (phases a and b) [2, 12]. Therefore, technical problems have often been applied in recent empirical research on problem-solving in pre- and primary school children [13–15].

Since problem-solving relies on various higher-order cognitive processes [7, 16], the requirements for age-appropriate measures assessing problem-solving skills and strategies in children are high [17]. The design cognition framework focusses on the cognitive processes involved in technical designing and problem-solving and is commonly applied in pre- and primary school research [15]. Design cognition research identifies the problem-solving phases based on participants' think-aloud utterances during design task performance (e.g., "Design and build a bug box that does not allow frogs in but allows bugs in/out" [15]). Thus, conclusions on cognitive processes during technical problem-solving are commonly drawn from protocol-coding of children's actions and their own verbal reasoning on them [15].

To examine technical problem-solving skills in preschool children, previous studies applied gear turning tasks (*GTT*) with toy-like gears, propellers and a plastic pegboard [18]. In these tasks, children were to assemble gears and propellers according to specific instructions in terms of their turning direction and turning speed (e.g., assembling gears so that they would turn in the same direction). In contrast to protocol-based evaluation methods, this study design provided the advantage that correct target states were clearly defined and the phases of problem-solving could be observed by analyzing the way participants organized, moved, interconnected and turned the gears and propellers [18]. Moreover, it integrated relevant scientific knowledge with logical understanding and goal-directed behavior. However, the analog implementation of the GTTs required considerable time in terms of set-up, coding and data processing.

Compared to analog testing, digital assessment methods offer numerous advantages [19]. Performance data, such as reaction times, can be saved more precisely and reliably. Moreover, coding algorithms can process data efficiently, rendering the need for the very time-consuming analog coding obsolete. Only a terminal device is needed to perform the experiment, instead of large quantities of analog materials. Instructions can be presented more standardized and their timing is more comparable across subjects, while the tasks can be designed in a child-friendly way [20]. In current research, digital measurement is increasingly preferred over analog measurement. For instance, formerly paper-based tests are successively digitized in neuropsychological contexts to enable more efficient diagnostic assessments [21]. Similarly, problem-solving skills in adolescents are assessed by means of digital paradigms, such as *microworlds*, simulating real-life problem situations [22, 23], and classic cognitive tests such as the Corsi Block-Tapping task, the Stroop task and the Trail-Making test are routinely administered digitally across a wide range of ages [24].

However, newly digitized test instruments require thorough validation [24], which is usually achieved by demonstrating convergent validity of the digital test against the analog counterpart [25]. Convergent validity of digital multiple-choice tests is usually shown by significant medium-sized correlations with the paper-and-pencil version [26]. When measuring more complex digitized tests assessing higher-order cognitive skills, like the Trail-Making test [27],

evidence for convergent validity is additionally provided by testing the strength of associations between related cognitive measures and both the digital and the analog versions of the test [28]. Moreover, the convergent validity is demonstrated via structural equation modelling assessing measurement invariance across latent factors of the analog and digital tests [29].

In this study, we developed digital, tablet-based versions of technical problem-solving tasks for children. We chose the GTTs that were applied in previous studies using analog test materials because they are appropriate for assessing five- to eight-year-old children's technical problem-solving [18]. Our aim was to validate the newly developed digital GTTs in six- to eight-year-old children. We therefore tested (1) whether performances on both modalities (analog and digital) were correlated with each other and with related measures (a language task, a reasoning task and another problem-solving task), (2) whether performance was measurement invariant across modalities, (3) whether performances on both modalities were predicted by related measures and (4) whether these predictive values differed between modalities. We expected substantial correlations between modalities, significant predictive values of related measures, that did not differ between modalities and that performances on both modalities (i.e., both task versions) were measurement invariant at the scalar level.

## Materials and methods

### Participants

A power-analysis (parameters: $r = .25$, $\alpha = .05$, $1\text{-}\beta = .90$) for correlational analyses resulted in a required sample size of $n = 164$. Power-analyses (parameters: $r = .25$, $\alpha = .05$, $1\text{-}\beta = .90$) for structural equation modeling analyses resulted in a required sample size of $n = 43$ for the highest degrees of freedom ($df = 7$) and $n = 54$ for the lowest degrees of freedom ($df = 5$). In sum, 215 children between six and eight years of age ($M = 7.18$ years, $SD = 0.78$; 89 female) participated in the study voluntarily and with written informed consent of their parents or caregivers. The recruitment period started on October 11, 2021 and ended on January 20, 2022. The study was approved by the local ethics committee (application #361). Participants were recruited in a town in southwestern Germany with a heterogeneous and diverse population [30]. The inclusion criterion was age (6–8 years; established prior to data analysis). We established no further exclusion criteria in order to minimize any selectivity of the sample.

### Procedure

Participants completed ten tasks across two 30-minute sessions with a 60-minute break. During both sessions, the child worked individually with a trained experimenter at the lab. In session one, children completed a language task, a reasoning task and a problem-solving task (*stabilization task*) to assess the convergent validity of the newly developed digital problem-solving tasks. Children also completed three conceptual knowledge tasks that are not relevant for the present analyses. Session two served to validate the new digital version of the problem-solving gear turning tasks (GTTs) against the analog version. It therefore included both GTTs (*carousel* and *propeller*) in both modalities (analog, digital). Task modality (analog, digital) and task type (carousel, propeller) were counterbalanced across participants. Sessions were video recorded and tasks were instructed in German language.

### Materials

All digital tasks were administered on a 10.1-inch sized tablet (Samsung Galaxy Note 10.1, Android version 5.1.1). The tablet program was developed in *Unity* (2020.3.17f1) and built via Android SDK (compression method LZ4). The digital GTTs were processed by single-point

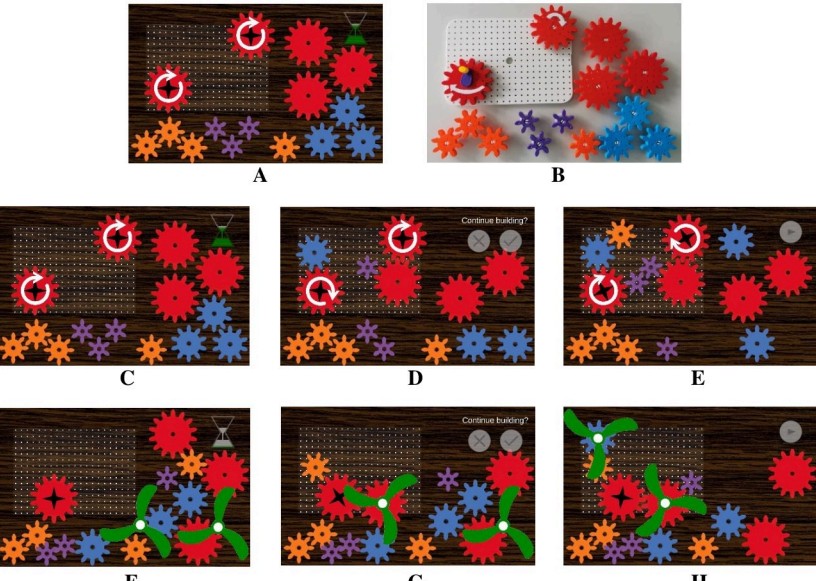

**Fig 1. Experimental set-up of the GTTs.** (A) Experimental set-up of the digital carousel task, (B) experimental set-up of the analog carousel task, (C) experimental phase 1 of the carousel task, (D) experimental phase 2 of the carousel task, (E) experimental phase 3 of the carousel task, (F) experimental phase 1 of the propeller task, (G) experimental phase 2 of the propeller task, (H) experimental phase 3 of the propeller task. GTT = Gear turning task.

swipe control in a two-dimensional response space (Fig 1A). Each gear had an inner and an outer radius. The inner radius was defined as the distance between a gear's center and the closest point of the notches between two gear teeth. The outer radius extended to the outer end of the gear teeth. The gears and propellers were moved via drag-and-drop with the currently selected object slightly protruding in perspective. When a gear was dropped onto the board, its new position was aligned with a grid that exactly mimicked the analog pegboard (20x14 plug-in options). Since the inner radii of two gears on the board could not overlap, gears released at an invalid position snapped in at the position with the smallest possible distance to the release position that was not less than the sum of the two overlapping inner gear radii. Gears located on the board could be turned by circularly swiping in the area between the inner and the outer radius. Whenever the distance between two gears was less than the sum of their outer radii and greater than the sum of their inner radii, they were considered to be connected and, thus, drove each other when turning. Propellers were dragged to the center of a gear to attach them to it and consequently moved and turned uniformly with this gear. Children had sufficient time to familiarize with the tablet handling. The analog version of the GTTs, carousel and propeller, was administered with a pegboard and plastic gears (Fig 1B).

## Tasks

**Language: Receptive vocabulary (Wechsler Preschool and Primary Scale of Intelligence (WPPSI-IV)) [31].** Participants saw consecutive displays with four pictures on a tablet screen and had to point to the picture named by the experimenter. The task included two practice trials and up to 35 test trials. It was aborted after three consecutive errors. The dependent variable was the number of correctly solved items.

**Reasoning: Matrix reasoning (WPPSI-IV [31]).** Participants saw a 2x2 matrix on a tablet screen containing three pictures and one question mark. They were instructed to pick one out

of four or five figurative response options to complete the matrix pattern. The task included three practice trials and up to 26 test trials. It was aborted after three consecutive errors. The dependent variable was the number of correctly solved items.

**Problem-solving: Stabilization task (cf. [32]).** Participants saw instable constructions of rectangular and triangular wooden blocks. Participants had up to three attempts to place another color-coded block stabilizing the construction. The task included one practice trial and eight test trials. The dependent variable was scored according to the number of required stabilizing attempts (items successfully solved at the first attempt: three points; one point less for each failed attempt; scoring range: 0–24).

**Problem-solving: Carousel task [18].** Participants saw a rectangular board with a driving gear and a target gear (introduced as a carousel) on it (Fig 1C). Both were marked by a circular arrow indicating a clockwise turning direction. Participants had to make the target gear turn clockwise when turning the driving gear clockwise by using other gears of four different sizes to connect driving and target gear. After three minutes or when participants indicated that they had finished constructing, the first experimental phase was over. In the second experimental phase, the experimenter removed the pegboard (analog modality) or the tablet (digital modality) temporarily and showed a printed picture of the initial state of the board to the participants (i.e., the state shown in Fig 1C). Subsequently, participants were asked to repeat the task requirements. If they were not able to repeat them correctly, the experimenter repeated them. The accuracy of the participants' responses was coded video-based, but not analyzed in this study. Afterwards, participants saw their construction and decided whether they wanted to make further changes to it or to end the task if they felt that it was completed. If they continued, they had up to two minutes to finish the construction (third experimental phase; Fig 1E). The dependent variable was the solution quality of the final construction state (scoring system: no gear was connected to either of the fixed gears (driving and target gear): 0 points; one of the fixed gears was connected to at least one other gear: 1 point; both fixed gears were connected to at least one other gear: 2 points; the two fixed gears were connected, but did not turn in the same direction: 3 points; they turned in the same direction: 4 points).

**Problem-solving: Propeller task [16].** Participants saw a rectangular board with an unmarked driving gear on it. In addition to different-sized gears, two propellers were provided (Fig 1F). Children had to attach them to gears in a way that one propeller would turn as fast as possible and the other one as slow as possible without touching each other when the driving gear was turned. Subsequently, the second and third experimental phase (shown in Fig 1G and 1H) proceeded in the same way as in the carousel task. The dependent variable was the solution quality of the final construction state. It was defined as the sum of two variables: *turning speed* (scoring system: at least one propeller was not attached to any gear: 0 points; both propellers were attached to the same gear type (i.e., gears of the same size): 1 point; both propellers were attached to different gear types, but not largest and smallest: 2 points; propellers were attached to the largest and smallest gear type: 3 points) and *contact* (scoring system: at least one propeller was not driven by the driving gear: 0 points; propellers touched when turning the driving gear: 1 point; they did not touch: 2 points).

## Analyses

Performance on the GTTs was evaluated by two trained raters coding the solution quality by rating the same dependent measures that were collected and calculated by the tablet program in the digital version. The mean inter-rater reliability between the raters for data of 20 randomly selected participants was very good for each dependent variable (Cohen's Kappa = .82-.97). Given that the correlation between the manually coded data of the digital GTTs and the

data measured by the tablet app was very substantial (carousel: $r = .87$, $p = .00$; propeller: $r = .94$, $p = .00$), we calculated the analyses based on the data measured by the tablet app.

Data of the analog GTTs of twelve participants was missing due to missing video recordings. Data of the analog propeller task of 21 participants was excluded because at least one propeller was not attached to the center but to the edge of a gear rendering the coding of turning speed impossible.

Statistical analyses were performed using R version 4.3.0 [33]. To analyze the manifest association between performances on both GTT modalities and other tasks, we used Spearman rank correlation analyses. As the main analysis, we estimated a model comprising two latent modality factors (analog and digital), each represented by performances on both GTTs (carousel and propeller task) of the respective modality. We fitted the covariance-based structural equation model with the R package lavaan [34] using only complete data sets. We evaluated goodness of fit based on the comparative fit index (*CFI*) [35] and the root mean square error of approximation (*RMSEA*) [36]. According to Hu and Bentler (1999), we considered CFI values >0.95 and RMSEA values <0.06 to indicate good model fit [37]. Additionally, we evaluated the standardized root mean square residual (*SRMR*) and the Tucker-Lewis index (*TLI*). According to Vermeent et al. (2022), we assessed longitudinal measurement invariance to control for configural (equal factor structure), metric (equal factor loadings) and scalar (equal intercepts) invariance across modalities using the likelihood ratio test [29]. Additionally, we included regression paths from the manifest indicators of language ability, reasoning ability and stabilization task performance to both modality factors in order to test their predictive value for GTT performance (convergent validity). We then imposed equality constraints on the regression paths in order to check for differences in this predictive value between modalities.

## Results

Descriptives and correlation coefficients for the dependent variables are presented in Table 1.

### Correlations between GTT modalities and other tasks

On the manifest level, we found significant associations between modalities in both tasks (carousel: $r = .33$, $p = .00$; propeller: $r = .50$, $p = .00$). Performance across all four GTTs correlated significantly with performance on the language task (carousel analog: $r = .31$; carousel digital: $r$

**Table 1. Means and standard deviations of the GTTs (carousel, propeller) and the validation tasks (reasoning, language and problem-solving) and correlations between these measures.**

| Variable | *M* | *SD* | 1 | 2 | 3 | 4 | 5 | 6 |
|---|---|---|---|---|---|---|---|---|
| 1. Language task | 27.02 | 4.44 | | | | | | |
| 2. Reasoning task | 16.20 | 4.47 | .41** | | | | | |
| 3. Problem-solving task | 12.22 | 3.20 | .37** | .36** | | | | |
| 4. GTT: Carousel analog | 3.35 | 1.05 | .31** | .31** | .32** | | | |
| 5. GTT: Carousel digital | 3.06 | 1.18 | .34** | .43** | .28** | .33** | | |
| 6. GTT: Propeller analog | 3.15 | 1.57 | .20** | .24** | .42** | .35** | .28** | |
| 7. GTT: Propeller digital | 2.43 | 1.66 | .38** | .34** | .36** | .24** | .46** | .50** |

Note: GTT = Gear turning task;

*$p < .05$,

**$p < .01$.

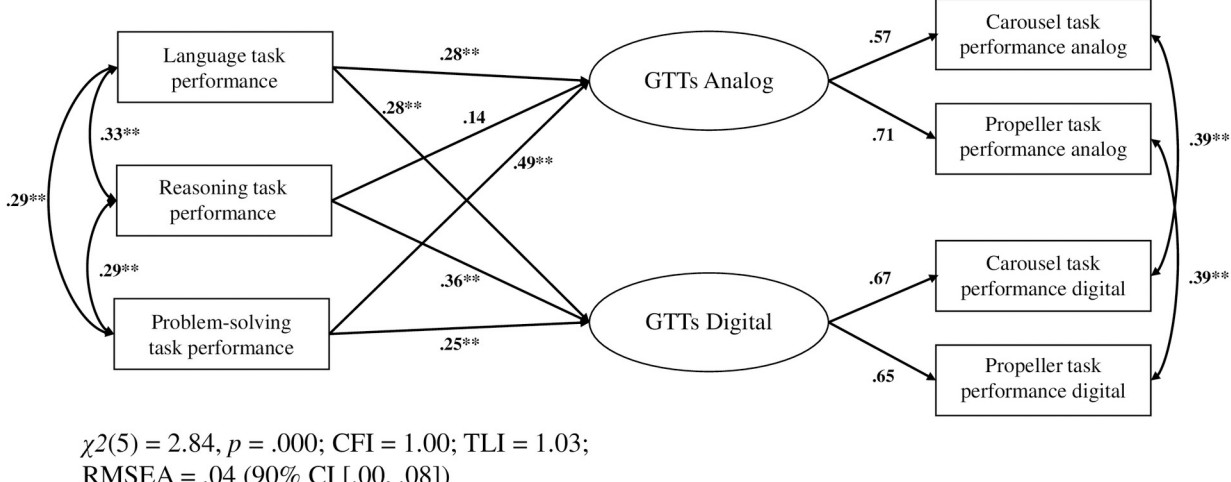

$\chi2(5) = 2.84$, $p = .000$; CFI = 1.00; TLI = 1.03;
RMSEA = .04 (90% CI [.00, .08])

**Fig 2. Model describing the relationships of analog and digital GTT performances and related measures (language, reasoning and stabilization problem-solving).** The squares represent manifest variables and the circles represent latent variables. Single-headed arrows from manifest variables to latent variables indicate regression paths, single-headed arrows from latent variables to manifest variables indicate factor loadings, double-headed arrows indicate correlations. The model was estimated with weighted least square mean and variance adjusted (WLSMV) for ordinal data. GTT = Gear turning task; *$p < .05$, **$p < .01$.

= .34; propeller analog: $r = .20$; propeller digital: $r = .38$; all $p < .01$). Performance on all GTTs was also significantly correlated with performance on the reasoning task (carousel analog: $r = .31$; carousel digital: $r = .43$; propeller analog: $r = .24$; propeller digital: $r = .34$; all $p < .01$). Moreover, performance on the GTTs was significantly correlated with performance on the stabilization task (carousel analog: $r = .32$; carousel digital: $r = .28$; propeller analog: $r = .42$; propeller digital: $r = .36$; all $p < .01$).

## Measurement invariance across GTT modalities

The latent factor model provided an excellent fit to the data (see Fig 2). Likelihood ratio tests revealed configural, metric and scalar measurement invariance across the modality factors (analog and digital; see Table 2).

## Predictive value of performance on related tasks for GTT performance (convergent validity)

Regression paths in the model (see Fig 2) were significant for language and stabilization performance across modalities. Reasoning was a significant predictor of digital GTT performance,

**Table 2. Model parameters of measurement invariance tests.**

| Level | $\chi2(df)$ | $p$ | Robust CFI | Robust TLI | Robust RMSEA [90% CI] | SRMR | $\Delta\chi2$ | $p(\Delta\chi2)$ |
|---|---|---|---|---|---|---|---|---|
| Configural | 2.839 (5) | .000 | 1.000 | 1.027 | .038 [.000-.082] | .027 | | |
| Metric | 3.449 (6) | .000 | 1.000 | 1.027 | .034 [.000-.080] | .032 | 0.610 | .326 |
| Scalar | 3.890 (7) | .000 | 1.000 | 1.028 | .033 [.000-.075] | .032 | 0.441 | .284 |

Note: Models were estimated with weighted least square mean and variance adjusted (WLSMV) for ordinal data. CFI = Comparative Fit Index; TLI = Tucker-Lewis Index; RMSEA = Root Mean Square Error of Approximation; CI = Confidence interval; SRMR = Standardized Root Mean Square Residual.

**Table 3. Comparison of regression paths between task modalities (analog, digital).**

| Model | χ2(df) | p | Robust CFI | Robust TLI | Robust RMSEA [90% CI] | SRMR | Δχ2 | p(Δχ2) |
|---|---|---|---|---|---|---|---|---|
| Unconstrained | 2.839 (5) | .000 | 1.000 | 1.027 | .038 [.000-.082] | .027 | | |
| Language | 2.972 (6) | .000 | 1.000 | 1.032 | .033 [.000-.074] | .028 | 0.133 | .480 |
| Reasoning | 5.365 (6) | .000 | 1.000 | 1.007 | .055 [.012-.096] | .039 | 2.526 | .047* |
| Stabilization task | 3.990 (6) | .000 | 1.000 | 1.021 | .041 [.000-.085] | .032 | 1.151 | .190 |

Note: All three regression paths were constrained separately to be equal across task modalities. Models were estimated with weighted least square mean and variance adjusted (WLSMV) for ordinal data. CFI = Comparative Fit Index; TLI = Tucker-Lewis Index; RMSEA = Root Mean Square Error of Approximation; CI = Confidence interval; SRMR = Standardized Root Mean Square Residual.

*$p < .05$,

**$p < .01$.

but not of analog GTT performance. Imposing equality constraints on the regression paths between modalities did not significantly change the model fit for language and stabilization performance, but for reasoning (see Table 3). Thus, language and stabilization performance contributed equally strong to GTT performances across modalities, whereas reasoning contributed more strongly to digital than to analog GTT performance.

## Discussion

Our aim was to validate new, digital versions of gear-based problem-solving tasks against the traditional analog tasks. Six- to eight-year-old children performed both the analog and the digital version of two GTTs as well as a number of related validation measures. First, we compared performance between both modalities (digital, analog). As expected, we found significant correlations between task performances across modalities. These results are consistent with findings from earlier studies [e.g., 24, 27, 38] that showed medium-sized correlations between digitized and analog versions of cognitive tests, indicating that both versions are comparable. Second, we found measurement invariance across the digital and the analog GTT versions on the configural, metric and scalar level. This indicates that both task versions had an equal factor structure, equal factor loadings and an equal scale level (i.e., equal intercepts), which also provides evidence for convergent validity. Third, we found the expected predictive value of language, reasoning and a problem-solving task performance for GTT performances on both modalities, providing further evidence for convergent validity of the tasks. Fourth, this predictive value was comparable across task modalities for language and stabilization performance, but larger in the digital modality for reasoning. The association between GTT performance and language ability indicates that children substantially relied on verbal processes to solve the GTT problems. In fact, this is in line with the assumption that children use subvocal verbal self-instruction to solve complex tasks [39–42]. The substantial association between GTT performance and stabilization task performance indicates that the new tasks are construct valid and might be representative for multiple problem-solving domains [43]. However, reasoning contributed more strongly to the digital than to the analog GTT performance. Since previous studies have shown that reasoning is an essential component of problem-solving [22, 44], this positive association confirmed the construct validity of the new digital measurement. The fact that reasoning was not predictive for the analog GTT performance might have been caused by a modality effect, since the reasoning test was also assessed digitally on a tablet.

The many advantages of digital testing [45] include that it can provide more accurate and detailed data and record multiple measurements without the need for trained personnel to perform video-based data coding (also eliminating the need to film hours of testing and get

parental consent for video recordings, increasing the efficiency of testing). The reliability of the digital data processing is clearly documented by very high correlations between human-coded data and data measured by the app. Furthermore, digital assessment methods are much easier to implement outside of the lab (e.g., in schools or daycare centers) and reduce test administrator influence on the assessment results [46]. Although the benefits of digital problem-solving assessment have been known for a long time [19], so far only very few digital assessment tools for children are available.

Regarding the framework of problem-solving, our study focused on solution quality rather than on the individual problem-solving phases [2]. Many previous studies lacked valid quantitative measures of solution quality, which may be of particular importance for the design of intervention studies and educational support programs. The digital GTTs automatically track both the solution quality as well as data providing information on the process of problem-solving, including the number and timing of gear turnings, gear displacements and processing time. This process data provides opportunities to closer investigate the cognitive processes (e.g., reasoning processes) involved in the different problem-solving phases. Therefore, the new tasks provide a significant improvement in the measurement of technical problem-solving strategies beyond qualitative approaches [15, 18] and extend current quantitative accuracy-based research on children's problem-solving [22, 23]. The digital GTTs could furthermore be adapted for the assessment of older children and adolescents by adjusting the difficulty level of the tasks, for example, by increasing distances between fixed gears or reducing the number of available gears. A limitation of our study is the lack of more specific information about the sample, such as the socio-economic status of the participants' caregivers.

## Conclusion

We conclude that the new digital GTTs are a valid adaptation of the traditional analog tasks for assessing problem-solving in six- to eight-year-old children. Given the substantial advantages of digital assessment instruments, the development of this tablet app contributes to research on cognitive development in problem-solving skills and strategies in early and middle childhood. The new tablet-based tasks also provide opportunities to investigate the contribution of higher-order cognitive abilities, such as executive functions, to problem-solving abilities in childhood [47]. Further studies are needed to collect representative normative data for the digital GTTs considering, for example, the effects of age, grade and gender.

## Acknowledgments

We especially want to thank Steffen Pohl and Rebekka Morath for their valuable assistance in the preparation of the study. Furthermore, we would like to thank the student research assistants for their support in data collection.

## Author Contributions

**Conceptualization:** Jonas Schäfer, Timo Reuter, Miriam Leuchter, Julia Karbach.

**Data curation:** Jonas Schäfer.

**Formal analysis:** Jonas Schäfer.

**Funding acquisition:** Miriam Leuchter, Julia Karbach.

**Investigation:** Jonas Schäfer, Timo Reuter.

**Methodology:** Jonas Schäfer, Julia Karbach.

**Project administration:** Jonas Schäfer.

**Resources:** Miriam Leuchter, Julia Karbach.

**Software:** Jonas Schäfer.

**Supervision:** Julia Karbach.

**Validation:** Jonas Schäfer.

**Visualization:** Jonas Schäfer.

**Writing – original draft:** Jonas Schäfer.

**Writing – review & editing:** Jonas Schäfer, Timo Reuter, Miriam Leuchter, Julia Karbach.

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
