## [Decision Letter · Decision Letter 0]

9 May 2024

PONE-D-24-09867Validation of new Tablet-based Problem-Solving Tasks in Primary School StudentsPLOS ONE

Dear Dr. Schäfer,

Thank you for submitting your manuscript to PLOS ONE. After careful consideration, we feel that it has merit but does not fully meet PLOS ONE’s publication criteria as it currently stands. Therefore, we invite you to submit a revised version of the manuscript that addresses the points raised during the review process.

**ACADEMIC EDITOR:**The study is technically sound and has major impact for an evolving field combining computer science and behavioral skills in young pupils.

Please address all the comments raised by the reviewers whom recommended minor revision. In particular language editing (proof-read by a native English speaker) and clarify your choice on the population (homogeneity).

Good luck and good continuation in this field.

We look forward to receiving your revised manuscript.

Kind regards,

Adel Tekari, PhD

Academic Editor

PLOS ONE

“This work was supported by the Ministry of Science, Further Education, Research and Culture, Rhineland-Palatinate, Germany. The funders awarded ML and JK (without grant number) and did not influence study or publication decisions.

https://bm.rlp.de”

Reviewers' comments:

Reviewer's Responses to Questions

**Comments to the Author**

1. Is the manuscript technically sound, and do the data support the conclusions?

Reviewer #1: Yes

Reviewer #2: Yes

2. Has the statistical analysis been performed appropriately and rigorously? 

Reviewer #1: Yes

Reviewer #2: Yes

3. Have the authors made all data underlying the findings in their manuscript fully available?

Reviewer #1: Yes

Reviewer #2: Yes

4. Is the manuscript presented in an intelligible fashion and written in standard English?

Reviewer #1: Yes

Reviewer #2: Yes

5. Review Comments to the Author

Reviewer #1: The authors present an approach for evaluating technical problem solving abilities in 6-8 years old elementary school children. The test, available in analog form, has been transferred to a tablet application, and the paper compares the two applications and related cognitive tasks (language, reasoning and another problem solving task), and the 2 forms of the problem solving test .

215 children voluntarily took part in the evaluation. The statistical methods for comparing the various tests are correlation studies and structural equation modeling. The results suggest that transfer of the problem solving task from analog to digital was successful .

The paper is well structured, understandably written, cites abundant relevant literature and includes clear result tables and figures. There are 3 comments that should be addressed:

the text - although understandable - should be revised by a native speaker or a proof reading service because some expressions and wordings could be improved. Suggestions can be given.

it is not clear, why only voluntary pupils participated in the study because this procedure could produce a selection bias, more technically gifted and computer affine children would be more likely to comprehend the computerized task than less technically gifted pupils. This could lead to the erroneous conclusion that less computer affine children could less successfully comprehend the tablet task and thus contribute to the erroneous conclusion that both tests may be used interchangeably, thus influencing its validity for all school children of the given age range. This could have been avoided by selecting a less homogeneous group. I would therefore suggest to include this possible bias in the limitations statement.

the third objection is that the order of test application was the same for all tests, analog before tablet. This could also lead to a selection bias because the analog task could be a training task thus facilitating the reasoning abut the given problem. This should also be mentioned in the limitation studies.

Reviewer #2: Although the benefits of digital problem-solving assessment have been known for a long time ,

so far ,only very few digital assessment tools for children are available. This work would therefore be of interest and would have an impact and a beneficial effect on the digital assessment of problem-solving skills in children aged 06 to 08.

the conclusion answers the questions and meets the objective of the study

the methodology is rigorous, with good statistical analysis

I do have a few comments on the methodology: I wonder why you chose not to give more information on the study population: gender, school level, development, management of emotions etc .. knowing that all these factors can influence problem solving.

So we don't know what type of population you've chosen and whether it's homogeneous or not

you also chose to make correlations with language and reasoning ability, but we also know that non-verbal intelligence ,self regulation and executive functions have a major influence on problem-solving skills.

have you taken these parameters into account?

if not, try to mention it during your discussion and in your conclusion and recommendations.

best regards

6. PLOS authors have the option to publish the peer review history of their article (what does this mean?). If published, this will include your full peer review and any attached files.

Reviewer #1: **Yes: **Christian Popow, MD

Reviewer #2: **Yes: **Hela Slama

---

## [Author Response · Author response to Decision Letter 0]

4 Jun 2024

R 1-1 the text - although understandable - should be revised by a native speaker or a proof reading service because some expressions and wordings could be improved. Suggestions can be given.

Response: We revised the entire manuscript carefully and improved language and wording.

Thank you for pointing this out!

R 1-2 it is not clear, why only voluntary pupils participated in the study because this procedure could produce a selection bias, more technically gifted and computer affine children would be more likely to comprehend the computerized task than less technically gifted pupils. This could lead to the erroneous conclusion that less computer affine children could less successfully comprehend the tablet task and thus contribute to the erroneous conclusion that both tests may be used interchangeably, thus influencing its validity for all school children of the given age range. This could have been avoided by selecting a less homogeneous group. I would therefore suggest to include this possible bias in the limitations statement. 

Response: In this context, voluntary refers to the fact that the participants actively agreed to participate in the study, which is mandatory for ethical reasons. However, we understand your concern regarding the selectivity of the sample. In order to recruit a heterogeneous sample, the study was advertised at a number of regional schools and was open to any child aged between six and eight years (no exclusion criteria). Experience with computers and other technology was not required and individuals from any educational and socioeconomic background were invited to participate.

We pointed this out clearer now:

“We established no further exclusion criteria in order to minimize any selectivity of the sample.” (p. 6)

R 1-3 the third objection is that the order of test application was the same for all tests, analog before tablet. This could also lead to a selection bias because the analog task could be a training task thus facilitating the reasoning abut the given problem. This should also be mentioned in the limitation studies. 

Response: In fact, we counterbalanced the task modality (analog vs. digital) across participants. This means that half of the participants started with the analog modality and the other half with the digital version to avoid the potential training bias that you described. To point this out more clearly, we extended the statement on p. 7:

“Task modality (analog, digital) and task type (carousel, propeller) were counterbalanced across participants.”

Thank you!

R 2-1 I do have a few comments on the methodology: I wonder why you chose not to give more information on the study population: gender, school level, development, management of emotions etc .. knowing that all these factors can influence problem solving. So we don't know what type of population you've chosen and whether it's homogeneous or not

Response: We only recorded age and gender of the participants (see p. 6). However, individuals of any educational and socioeconomic background were invited to participate to minimize any selectivity of the sample. Also, participants were recruited in a town in Southwestern Germany with a statistically highly heterogeneous and diverse population.*

We have expanded our methods section accordingly:

“We established no further exclusion criteria in order to minimize any selectivity of the sample" (p. 6)

However, we acknowledge that more information on the sample would have been helpful and added this point as a limitation to the discussion: 

“A limitation of our study is the lack of more specific information about the sample, such as the socio-economic status of the participants’ caregivers.“ (p. 16)

*Ackel-Eisnach, K. (2023). Landau in Zahlen – Analyse bevölkerungsstatistischer Indikatoren zur gesellschaftlichen Vielfalt. In: Pusch, B., Spieker, S., Horne, C. (eds) Heterogenität und Diversität in Städten mittlerer Größe. Springer VS, Wiesbaden. https://doi.org/10.1007/978-3-658-39076-1_2

R 2-2 you also chose to make correlations with language and reasoning ability, but we also know that non-verbal intelligence ,self regulation and executive functions have a major influence on problem-solving skills. have you taken these parameters into account? if not, try to mention it during your discussion and in your conclusion and recommendations. 

Response: The main goal of this study was to validate the newly developed digital tasks. As a first step, we therefore focused on testing their convergent and divergent validity based on a selected battery of cognitive tasks. However, we completely agree that nonverbal intelligence, self-regulation and executive functions are important correlates of problem-solving ability. While reasoning task applied in this study assessed an aspect of non-verbal intelligence, we did not have the opportunity to include measures of self-regulation and executive functions in this study (because of time limitations during data collection). Therefore, we assessed the correlation between performance on the problem-solving tasks and measures of self-regulation and executive functions in a second study that has been published elsewhere (Schäfer et al., in press). 

In order to acknowledge your suggestion, we added a statement in the Conclusion section, which refers to our follow-up research:

“The new tablet-based tasks also provide opportunities to investigate the contribution of higher-order cognitive abilities, such as executive functions, to problem-solving abilities in childhood (Schäfer et al., in press).” (p. 16)

Consequently, we added the following reference to the reference list:

47. Schäfer, J., Reuter, T., Leuchter, M., & Karbach, J. (in press). Executive functions and problem-solving – the contribution of inhibition, working memory, and cognitive flexibility to science problem-solving performance in elementary school students. Journal of Experimental Child Psychology.

Thank you very much!

---

## [Decision Letter · Decision Letter 1]

22 Jul 2024

PONE-D-24-09867R1Validation of new Tablet-based Problem-Solving Tasks in Primary School StudentsPLOS ONE

Dear Dr. Schäfer,

Thank you for submitting your manuscript to PLOS ONE. After careful consideration, we feel that it has merit but does not fully meet PLOS ONE’s publication criteria as it currently stands. Therefore, we invite you to submit a revised version of the manuscript that addresses the points raised during the review process.

Please conduct some minor changes according to reviewer's comments.

Good luck.

We look forward to receiving your revised manuscript.

Kind regards,

Adel Tekari, PhD

Academic Editor

PLOS ONE

Journal Requirements:

Reviewers' comments:

Reviewer's Responses to Questions

**Comments to the Author**

1. If the authors have adequately addressed your comments raised in a previous round of review and you feel that this manuscript is now acceptable for publication, you may indicate that here to bypass the “Comments to the Author” section, enter your conflict of interest statement in the “Confidential to Editor” section, and submit your "Accept" recommendation.

Reviewer #2: All comments have been addressed

Reviewer #3: (No Response)

2. Is the manuscript technically sound, and do the data support the conclusions?

Reviewer #2: Yes

Reviewer #3: Yes

3. Has the statistical analysis been performed appropriately and rigorously? 

Reviewer #2: Yes

Reviewer #3: Yes

4. Have the authors made all data underlying the findings in their manuscript fully available?

Reviewer #2: Yes

Reviewer #3: Yes

5. Is the manuscript presented in an intelligible fashion and written in standard English?

Reviewer #2: Yes

Reviewer #3: Yes

6. Review Comments to the Author

Reviewer #2: Interesting and innovative research

thank you for the changes made and added to the paper

good luck

Reviewer #3: The authors provide sufficient evidence to validate a digital version of an analog problem-solving task for primary school children. The practical relevance of their contribution is contextualized in the current literature, pointing to the significant advantages of digital platforms compared to analog formats. The methodological approach was rigorous enough. Although I have some minor observations and suggestions mostly regarding methods, overall, I think that the manuscript meets the criteria to be published in Plos One.

My commentaries are:

- In Material and Methods, the description of the Language and Reasoning tasks (page 10 of the revised version) should indicate that they were applied in digital format. This is not deduced from the descriptions offered, and it matters for reproducibility and because the authors indeed allude to a difference in format to explain the lack of predictive relation between Reasoning and the analog GTT tasks.

- In the description of the Carrousel task (page 10), the rationale and steps of the second phase are not immediately clear. It is not so common in diagnostic tasks to include probing questions during the task itself (after having had a familiarization or a practice phase). I suggest including some indication about this (especially if the author's goal is that the community can use their tasks upon the information given in this manuscript). Also, if the authors deem it useful to code subject's responses on this phase to complement the assessment, this should be pointed out.

- In the Discussion, the authors suggest that the absence of a link between Reasoning and analog GTT performance could have been due to a difference in the tasks' formats (page 17 of the revised version). While I agree that this is plausible, it may lead to think then that the observed relation between Reasoning and digital GTT might have resulted from their shared format (and not because of an underlying cognition). As the reasoning process in the GGT tasks was not measured here (only the quality outcome) a suggestion for a future exploration of these intermediate steps and cognitive tasks may help.

- In the Participants section, as socioeconomic information was not registered, in addition to the statement of no exclusion criteria (page 6), I suggest adding that the sampling area had a heterogeneous population (to compensate for this limitation).

- Regarding software availability, it would be important to know if the authors plan to provide access to their developed tool to the community.

Minor regarding format:

- In the Results, the format to present correlation coefficients in the text, separating them by "/" should be checked (pages 13 and 14 of the revised version).

7. PLOS authors have the option to publish the peer review history of their article (what does this mean?). If published, this will include your full peer review and any attached files.

Reviewer #2: **Yes: **Hela Slama

Reviewer #3: **Yes: **Camilo Gouet

---

## [Author Response · Author response to Decision Letter 1]

25 Jul 2024

Comment

In Material and Methods, the description of the Language and Reasoning tasks (page 10 of the revised version) should indicate that they were applied in digital format. This is not deduced from the descriptions offered, and it matters for reproducibility and because the authors indeed allude to a difference in format to explain the lack of predictive relation between Reasoning and the analog GTT tasks. 

Response

We added this information to both task descriptions (p. 8 in the current manuscript):

Language: “Participants saw consecutive displays with four pictures on a tablet screen and had to point to the picture named by the experimenter.”

Reasoning: “Participants saw a 2x2 matrix on a tablet screen containing three pictures and one question mark.”

Thank you!

Comment

In the description of the Carrousel task (page 10), the rationale and steps of the second phase are not immediately clear. It is not so common in diagnostic tasks to include probing questions during the task itself (after having had a familiarization or a practice phase). I suggest including some indication about this (especially if the author's goal is that the community can use their tasks upon the information given in this manuscript). Also, if the authors deem it useful to code subject's responses on this phase to complement the assessment, this should be pointed out. 

Response

We agree that our description of the second phase of the carousel task was not entirely clear. We therefore rephrased the corresponding paragraph (p. 9):

“In the second experimental phase, the experimenter removed the pegboard (analog modality) or the tablet (digital modality) temporarily and showed a printed picture of the initial state of the board to the participants (i.e., the state shown in Fig 1C). Subsequently, participants were asked to repeat the task requirements.”

Regarding the second task phase, we coded how well the subject’s responses matched the actual task requirement. Since this study focused on outcome performance, we did not analyze this measure here. However, to make this point more transparent, we integrated the following statement in the task description (p. 9):

“The accuracy of the participants’ responses was coded video-based, but not analyzed in this study.”

Comment

In the Discussion, the authors suggest that the absence of a link between Reasoning and analog GTT performance could have been due to a difference in the tasks' formats (page 17 of the revised version). While I agree that this is plausible, it may lead to think then that the observed relation between Reasoning and digital GTT might have resulted from their shared format (and not because of an underlying cognition). As the reasoning process in the GGT tasks was not measured here (only the quality outcome) a suggestion for a future exploration of these intermediate steps and cognitive tasks may help. 

Response

This is a very interesting aspect. We added this aspect to the Discussion (p. 16):

“This process data provides opportunities to closer investigate the cognitive processes (e.g., reasoning processes) involved in the different problem-solving phases. Therefore, the new tasks provide a significant improvement in the measurement of technical problem-solving strategies beyond qualitative approaches [15, 19] and extend current quantitative accuracy-based research on children's problem-solving [23, 24].”

Thank you very much for this interesting insight!

Comment

In the Participants section, as socioeconomic information was not registered, in addition to the statement of no exclusion criteria (page 6), I suggest adding that the sampling area had a heterogeneous population (to compensate for this limitation). 

Response

We added the following statement in the Participants section to provide more information on the sample (p. 6):

“Participants were recruited in a town in southwestern Germany with a heterogeneous and diverse population [31].”

Comment

Regarding software availability, it would be important to know if the authors plan to provide access to their developed tool to the community. 

Response

We made this aspect explicit in the Data availability statement (p. 17):

“The software used in this study will be made available on request by e-mail to the corresponding author.”

Comment

In the Results, the format to present correlation coefficients in the text, separating them by "/" should be checked (pages 13 and 14 of the revised version).

Response

We revisited this formatting style to avoid backslashes and to make clearer which coefficient refers to which task modality.

Thank you very much for your effort!

---

## [Decision Letter · Decision Letter 2]

19 Aug 2024

Validation of new Tablet-based Problem-Solving Tasks in Primary School Students

PONE-D-24-09867R2

Dear Dr. Schäfer,

We’re pleased to inform you that your manuscript has been judged scientifically suitable for publication and will be formally accepted for publication once it meets all outstanding technical requirements.

Kind regards,

Adel Tekari, PhD

Academic Editor

PLOS ONE

Additional Editor Comments (optional):

The authors addressed all the reviewers' comments and no additional action is required. Congratulations for the paper.

Reviewers' comments:

Reviewer's Responses to Questions

**Comments to the Author**

1. If the authors have adequately addressed your comments raised in a previous round of review and you feel that this manuscript is now acceptable for publication, you may indicate that here to bypass the “Comments to the Author” section, enter your conflict of interest statement in the “Confidential to Editor” section, and submit your "Accept" recommendation.

Reviewer #3: All comments have been addressed

2. Is the manuscript technically sound, and do the data support the conclusions?

Reviewer #3: (No Response)

3. Has the statistical analysis been performed appropriately and rigorously? 

Reviewer #3: (No Response)

4. Have the authors made all data underlying the findings in their manuscript fully available?

Reviewer #3: (No Response)

5. Is the manuscript presented in an intelligible fashion and written in standard English?

Reviewer #3: (No Response)

6. Review Comments to the Author

Reviewer #3: (No Response)

7. PLOS authors have the option to publish the peer review history of their article (what does this mean?). If published, this will include your full peer review and any attached files.

Reviewer #3: **Yes: **Camilo Gouet

---

## [Editor Report · Acceptance letter]

21 Aug 2024

PONE-D-24-09867R2 

PLOS ONE

Dear Dr. Schäfer, 

I'm pleased to inform you that your manuscript has been deemed suitable for publication in PLOS ONE. Congratulations! Your manuscript is now being handed over to our production team.

Kind regards, 

on behalf of

Dr. Adel Tekari 

Academic Editor

PLOS ONE